# The Mutual Incorporation of Mg^2+^ and CO_3_^2−^ into Hydroxyapatite: A DFT Study

**DOI:** 10.3390/ma15249046

**Published:** 2022-12-17

**Authors:** Olga N. Makshakova, Marat R. Gafurov, Margarita A. Goldberg

**Affiliations:** 1Kazan Institute of Biochemistry and Biophysics, Federal Research Center Kazan Scientific Center of Russian Academy of Sciences, 420111 Kazan, Russia; 2Institute of Physics, Kazan Federal University, 420008 Kazan, Russia; 3A.A. Baikov Institute of Metallurgy and Materials Science, Russian Academy of Sciences, 119334 Moscow, Russia

**Keywords:** hydroxyapatite, co-doping, Mg substitution, carbonate substitution, density functional theory

## Abstract

Hydroxyapatite (HA) with a stoichiometry composition of Ca_10_(PO_4_)_6_(OH)_2_ is widely applied for various biomedical issues, first of all for bone defect substitution, as a catalyst, and as an adsorbent for soil and water purification. The incorporation of foreign ions changes the acid–base relation, microstructure, porosity, and other properties of the HA materials. Here, we report the results of calculations of the density functional theory and analyze the possibility of two foreign ions, CO_3_^2−^ and Mg^2+^, to be co-localized in the HA structure. The Na^+^ was taken into account for charge balance preservation. The analysis revealed the favorable incorporation of CO_3_^2−^ and Mg^2+^ as a complex when they interact with each other. The energy gain over the sole ion incorporation was pronounced when CO_3_^2−^ occupied the A position and Mg^2+^ was in the Ca(2) position and amounted to -0.31 eV. In the most energy-favorable complex, the distance between Mg^2+^ and the O atom of carbonate ion decreased compared to Mg…O distances to the surrounding phosphate or hydroxide ions, and amounted to 1.98 Å. The theoretical calculations agree well with the experimental data reported earlier. Understating the structure–properties relationship in HA materials varying in terms of composition, stoichiometry, and morphology paves the way to rational designs of efficient bio-based catalytic systems.

## 1. Introduction

### 1.1. Hydroxyapatite’s Applications

Hydroxyapatite (HA) is one of the most important bio-ceramics, which gains more attention in the form of porous scaffolds [1], granules [2], the component of composite materials [3,4,5], coatings [6], and nanoparticles [7,8]. Due to its derived surface with acid–base properties, HA has been widely used in organic synthesis and heterogeneous catalysis [9,10,11,12]. The wide range of reactions includes the formation of C–C, C–O, C–N, and C–S bonds, and reactions of oxidation, reduction, cycloadditions, etc. The stoichiometric composition of HA is Ca_10_(PO_4_)_6_(OH)_2_. At the stoichiometric Ca/P ratio, HA behaves as a basic catalyst despite the acid site presence. At a low non-stoichiometric ratio between Ca and P amounting to 1.50, HA behaves as an acid catalyst, despite the basic site presence [13].

The HA properties and morphology could be modified by tuning their composition and stoichiometry via the inclusion of ‘foreign’ ions [13,14]. Typical trace elements in HA of bones and tooth enamel are ~4–8 wt.% of carbonate; ~0.4–0.9 wt.% of sodium and magnesium; and others such as potassium, fluoride, and chloride are less than 0.1 wt.% [15]. Additionally, the catalytic properties could be improved by the introduction of different transitional metals such as Ni [16] or Fe [17]. The morphology, stability, solubility, mechanical properties, and biological behaviors of HA synthesized in vitro are also modulated by the incorporation of foreign ions. According to their size and charge, cations, such as Na^+^, Zn^2+^, Mg^2+^, and Fe^3+^ substitute Ca^2+^ [18,19,20,21]. In contrast, anions, such as CO_3_^2−^, SiO_4_^4−^, F^−^, and Cl^−^ can theoretically substitute PO_4_^3−^ (B position) or OH^−^ (A position) (Figure 1). It is important to note, the size of the anion is, apparently, decisive, i.e., comparably large SiO_4_^4-^ was reported to be found only in the B position [22,23,24].

### 1.2. Carbonate Substitutions

Since carbonated HA is a major component of bone and dental enamel, it has been extensively studied experimentally [25,26,27,28,29,30,31,32]. There are also numerous computational studies of the inclusion of CO_3_^2−^ into HA cells [33,34,35]. The in silico studies were conducted using the ab initio crystal-field method [33], density functional theory (DFT) [35], and molecular dynamics (MD) [34]. Both natural and synthetic samples present different concentrations of type A and type B defects; however, their distribution within the lattice is not clear [36]. The MD simulations revealed that upon the carbonate uptake from the solution into the HA crystal, the type A defect was energetically preferred, followed by type AB, and then type B [34]. The DFT calculations using generalized gradient approximation and Perdew–Burke–Ernzerhof (PBE) exchange-correlation potential showed that the most energetically stable substitution was rather type AB, followed by type A, and then type B [35].

The analysis by FTIR spectroscopy was capable of distinguishing the A and B positions of CO_3_^2−^ characteristic for bone and enamel: (1) apatite channel, oriented with two oxygen atoms close to the *c*-axis (type A1); (2) close to a sloping face of the PO_4_ tetrahedron (type B). The third position in a stuffed channel (type A2) was a feature of high-pressure [30]. There was a report claiming that type B substitution was more favorable than type A substitution, revealed by FTIR spectra and the computed crystal energy [37]. In our recent study, the trace amount of CO_3_^2−^ in synthesized HA was of the A-B substitutional type [38].

The introduction of CO_3_^2−^ requires a charge compensation. For the charge balance in type B carbonated HA, the Na^+^ co-substitution was experimentally identified as the most likely case [39]. Sodium ion was observed in biological apatites, and the Na^+^ radius was close to that of Ca^2+^ ion [36]. In contrast, Astala and Stott [40] found the most favorable charge compensation scheme for the B type by adding a hydrogen atom linked to a neighboring phosphate.

The values of the HA specific surface area (SSA) could be controlled by the CO_3_^2−^ anions’ introduction in the structure. Thus, as CO_3_^2−^ anions were introduced, the SSA value grew from 79 to 107 m^2^/g [41]. When Na^+^ and CO_3_^2−^ ions were added during the synthesis, the powders with the HA structure with a slight change in the *a* and *c* parameters were obtained. These materials were characterized by the presence of both acidic and basic properties, which could also be controlled by the variation of the Ca/P ratio. While the existence of HPO_4_^−^ ions led to the increase in acid sites, the addition of the CO_3_^2−^ anions resulted in the growth of the basicity of HAp. At the same time, the presence of Na^+^ resulted in the complementary basic sites’ increase. Thus, the obtaining of the Ca-deficient HA with the introduction of both Na^+^ and CO_3_^2−^ ions led to the enhancement and adjustment of the acidic and basic surface properties. The presented Na^+^ and CO_3_^2−^ co-doped HA could be applied as the efficient catalysts in the thiolation reaction of methanol [41].

### 1.3. Magnesium Substitutions

The Mg^2+^ contents in enamel, bone, and dentin are about 0.44, 0.72, and 1.23 wt.%, respectively [42,43]. It plays a key role in bone metabolism, in particular, during the early stages of osteogenesis and when magnesium-depleted bones are fragile [44,45]. When magnesium was introduced in the HA structure, the decrease in the degree of crystallinity, thermal stability, and strong distortions of the HA lattice was observed. This was due to the distinction of the ionic radius of Mg^2+^ (0.72 Å) compared to Ca^2+^ (1.04 Å) [46]. According to [47], the introduction of Mg^2+^ led to the growth of parameter *a* with the absence of the noticeable changes in parameter *c*, when Mg-HA mesoporous powders were obtained by the microwave-assistance synthesis [47]. Laurencin et al. [42] demonstrated that the introduction of Mg^2+^ resulted in a fall of parameter *a*, regardless of the substituted position. The magnesium and carbonated co-doped powders were characterized by a similar tendency when materials were synthesized by the wet synthesis after the drying at 60 °C [48].

According to the Rietveld and DFT ab-initio calculations, the magnesium cations could occupy the Ca(1) in the HA structure [46,49]. At the same time, the substitution Ca(2) position by the Mg^2+^ was supported by multinuclear solid-state NMR and X-ray absorption spectroscopy [42]. Upon Mg^2+^ incorporation, the environment of the anions was disordered. The calculations showed the shortening of Mg…O distances and modifications of the HA lattice [38,42].

The SSA of Mg^2+^-doped HA could be lower and higher compared to HA, depending on the synthesis route and magnesium amount [38,50,51]. The existence of magnesium cations, which were characterized by the more acidic sites, compared to calcium cations could increase the final yield of the Biginelli product. At the same time, the leaching of magnesium during the catalytic process is a challenge up to today, which has led to limitations of the application of MgHAp as a catalyst [13].

### 1.4. Magnesium and Carbonate Co-Substitutions

The incorporation of magnesium could facilitate the presence of carbonate. The carbonated Mg-substituted HA is closer to native human bone tissue. The type of carbonate substitution is linked with the synthesis route. Lala et al. [52] synthesized materials via the ball milling of the MgO, CaCO_3_, and CaHPO_4_·2H_2_O, and presented the preferential introduction of Mg^2+^ in the Ca(2) position for the type A carbonated HA according to the Rietveld’s refinement for XRD and FTIR analysis. The experiments on the HAp powders’ precipitation from the simulated body fluid solution showed that Mg^2+^ and CO_3_^2−^ ions coming together were favorably incorporated in the powders [53]. These may imply that the co-substitution of the ions is favorable in HAp [54,55] Our recent work revealed that the introduction of Mg^2+^ results in the more predictable formation of A type CO_3_^2−^, unlikely to carbonate HA where a mixed A-B type of CO_3_^2−^ incorporation occurs. Additionally, the type of carbonate substitution is affected by the heat treatment, and A is energetically more favorable in the high-temperature environment, whereas the B type is preferred to the A type in the aqueous solution environment.

In the current work, we present the DFT study of HA co-doped with CO_3_^2−^ and Mg^2+^. The substitution of divalent Ca^2+^ by Mg^2+^ was performed in both Ca(1) and Ca(2) positions. For the incorporation of CO_3_^2−^, the charge compensation was as follows: (1) when the carbonate ion was in the A position it replaced two OH^−^ ions [35], (2) when the carbonate ion was in the B position, one Ca^2+^ was replaced with Na^+^ [36]. Then, the combination of positions for all components was thoroughly analyzed. The energy gain upon the co-incorporation of CO_3_^2−^ and Mg^2+^ has been calculated again as the energy gain from the sole ion CO_3_^2−^ or Mg^2+^ incorporation into HA. The positions of the foreign ions’ incorporation and the scheme of charge compensation would crucially affect the local acid/base properties of doped HA and their use in catalysis. The results of the modelling would create a platform for the rational design of such materials and provide important parameters, as cell dimensions, to verify the ions’ incorporation at given experiment conditions.

## 2. Methods

The 1 × 1 × 1 monoclinic HA (88 atoms in the cell) was applied for calculations. The initial unit–cell parameters were taken from [56]. Previously, for both pure HA and HA containing foreign ions, it was shown that one monoclinic cell (space group P21/b) was sufficient to reproduce the g-factors and A-hyperfine values for various paramagnetic species [57,58] and phonon spectral density [59]. It was possible to adequately describe HA systems containing foreign cations, such as aluminum, iron, and magnesium cations [18,38,60,61]. It is important to note, the descriptions of monoclinic and hexagonal cells (usually found in the experiment) use different axis notation which should not lead to confusion, as demonstrated by our results [60].

The foreign ions were incorporated as follows: cations substituted Ca^2+^ in two positions, and carbonate anion substituted anions in a position of orthophosphate or hydroxide ions. To respect the principle of electroneutrality, the inclusion of divalent carbonate in the position of the OH^-^ group required an additional hydroxide ion elimination. Then, CO_3_^2−^ was incorporated into the PO_4_^3−^ position, and one of the neighboring calcium ions (in the Ca(1) or Ca(2) position) was replaced with a sodium cation.

The optimization of cell geometry was carried out at the DFT level. A plane-wave basis was used with the ultrasoft pseudopotentials of Vanderbilt [62]. The generalized-gradient approximation was used in the scheme of Perdew–Wang (GGA PW91) to describe the exchange–correlation functional. [63]. For the smooth part of the electron wave functions, the 45 Ry kinetic energy cutoff was taken. For the augmented electron density, the cutoff value was 300 Ry. All calculations were performed using the Quantum ESPRESSO program package [64].

In the course of geometry optimization, both atomic positions and cell geometry were allowed to change. The convergence on forces was achieved with the threshold value 10^−3^ a.u. The Monkhorst-Pack scheme was used to sample the Brillouin zone with 2 × 2 × 1 k-point mesh [65].

The energy change upon the incorporation of a pair of foreign ions was calculated as follows (1):ΔE = ΔE_HA+Mg+CO3_ − ΔE_HA+Mg_ − ΔE_HA+CO3_ + ΔE_HA_(1)
where ΔE_HA+Mg+CO3_ stands for the HA system containing both magnesium and carbonate ions; ΔE_HA+Mg_–HA containing magnesium foreign ion only; ΔE_HA+CO3_–HA containing carbonate foreign ion only; ΔE_HA_–the energy of pure HA.

## 3. Results and Discussion

### 3.1. Mg^2+^ Incorporation into HA

Divalent magnesium may substitute Ca^2+^ in HA without other compositional alternations since no charge compensation is needed. However, due to the large size difference (according to the Pauling scale, Mg^2+^ is about 0.28 Å smaller than Ca^2+^), the substitution leads to the contraction of the cell and distortion of the HA lattice [42]. At an atomistic level, it relates to a local shift of the ions coordinating Mg^2+^ [38].

In the cell of HA, there are two positions of Ca^2+^ calcium ions (Figure 1). In the first position, referred to as Ca(1), coordination of Ca^2+^ occurs via 6 oxygen atoms from 4 orthophosphates. The first coordination sphere of calcium in the Ca-2 position includes 7 oxygen atoms: 6 from four orthophosphates and 1 hydroxide. In the literature, the Mg^2+^ substitution was hypothesized in both Ca(1) [35,66] and Ca(2) positions [42,54]. The quantum chemical energy estimations at the GGA PBE level of theory revealed both the preference of the Ca(1) substitution by 0.012 eV [46] and Ca(2) substitution by 0.25 eV [42] and by ~0.1 eV [67]. Here, we present our own calculations to resolve the ambiguity. Furthermore, the energy and cell parameters of the Mg-doped HA cell are crucial to deduce the influence of carbonate ion upon their mutual incorporation into HA, which will be analyzed in the following chapter.

Some parameters of the cell geometry upon Mg^2+^ inclusion and relative values for pure HA are given in Table 1. According to the Ca…O distances in the first coordination shell, the surrounding of Ca(2) forms tighter contacts with the minimal distance to the calcium ion, amounting to 2.36Å (Figure 2). In the Ca(1) position, the shortest Ca…O distance is larger at 2.39 Å. The geometric parameters are close to those calculated at the GGA PBE level of theory [18]. When Mg^2+^ is incorporated into a calcium position, the first coordination shell overcomes a notable perturbation. The contraction in the first coordination shell results in the decrease in the Mg^2+^…O distances. In the Ca(1) position, the minimal distance between the magnesium ion and the closest oxygen atom of orthophosphate decreases to 2.15 Å. In the Ca(2) position, the closest oxygen atom (the hydroxide one) appears at a distance of 2.09 Å. One might anticipate that a more noticeable local condensation of ions would lead to a greater compression of the lattice cell. In contrast, despite the most pronounced shortening of the Mg^2+^…O distances in the Ca(2) position, the contraction of the cell lattice is more pronounced when Mg^2+^ is in the Ca(1) position. To form shorter contacts in the coordination sphere, Mg^2+^ that resides in the Ca(2) position is shifted towards phosphate groups by 0.29 Å from the related position of Ca^2+^. The OH^−^ ion is shifted towards Mg^2+^ by 0.3Å so that it declines from the axis of the anion channel. This distortion of the anion channel is supported by the FTIR data [38].

In addition, our results of energy estimation support the inclusion of Mg^2+^ into the Ca(2) position with a slight preference: the energy gain into Ca(2) against Ca(1) amounts to 0.042 eV (Table 1).

Despite the fact that the energy preference is not largely pronounced, our results are rather supportive of the results of the Laurencin et al. [42] and Saito et al. [67] reports, rather than the Ren et al. one [35].

### 3.2. Mg^2+^ and CO_3_^2−^ Co-Incorporation into HA

#### 3.2.1. Carbonate in the OH^−^ (A) position

When HA carbonated in the A position (Figure 3) and magnesium is incorporated into the position of the Ca(2), it has the most favorable energy, −0.308 eV (Table 2, Figure 4). This implies the incorporation of Mg^2+^ and CO_3_^2^**^−^** happens more beneficially as a complex than as sole ions. Indeed, the combination of the carbonate positioning in the anion channel and the magnesium positioning in the layer surrounding the anion channel allows the two ions to interact at the Mg…O distance of 1.98 Å. This distance is shorter than the Mg…O distances to orthophosphate or hydroxide ions, at least by 0.11 Å. In contrast, when Mg^2+^ is incorporated in the Ca(1) position, which is remote from the anion channel, the magnesium and carbonate ions cannot interact with each other. Such incorporation as sole ions does not lead to any energy change.

When CO_3_^2−^ incorporates in the A position, the cell slightly expands. The following incorporation of magnesium cation into the Ca(1) position results in an additive cell contraction. Interestingly, there are no cell size changes observed upon the following incorporation of magnesium into the Ca(2).

#### 3.2.2. Carbonate^−^ in the PO_4_^3−^ (B) position

The incorporation of CO_3_^2−^ into the B position requires a compositional alteration for the charge compensation. The most commonly used scheme is the substitution of one Ca^2+^ with one Na^+^ ion. This is due to their comparable size and the fact that sodium impurity is one of the most commonly present factors in natural HA, and simultaneously with carbonate [36]. The skeleton contains 99% of the calcium of the body, 35% of the body’s sodium, 80% of the body’s carbonates, and 60% of the magnesium [68], and additional trace elements such as K, Cr, Zn, and Sr [69]. The carbonate content of bone mineral is shown to vary depending on the age of the individual [70,71], and type B carbonate is the most abundant species in the bones of young humans, with increasing A type in the bones of old people [72]. Thus, in the current section, we explore the panel of possible combinations of the inclusion of both Na+ and Mg^2+^ in the Ca(1) and Ca(2) positions, keeping CO_3_^2−^ in the B position, similar to in the native human bone [73].

The incorporation of CO_3_^2−^ together with Na^+^ results in a contraction of the cell by 0.6–0.7%, irrespective of the Na^+^ position. The following incorporation of Mg^2+^ leads to further cell contraction by 0.7% and 1.0%, respectively, for the Ca(2) and Ca(1) substitution, which are in agreement with the result of Section 3.1.

All variations of the Mg^2+^ and Na^+^ positioning show the distances between Mg^2+^ and CO_3_^2−^ as long as in the Mg^2+^-doped HA, which is lacking CO_3_^2−^ in the B position. The mutual orientation of Mg^2+^ and the carbonate at such distances does not bring any notable energy gain upon the mutual incorporation of the ions to HA (Table 2). Tighter interactions with the carbonate ion are not possible because of forces acting on the Mg^2+^ from the surrounding ions in the cell. There is no additional impact from Na^+^ on the interactions between Mg^2+^ and CO_3_^2−^. Previously, the synthesis of Mg-doped carbonate-apatite containing 6 wt.% carbonate was performed by dropping sodium hydrogen carbonate (NaHCO_3_) solution and H_3_PO_4_ into the suspension of MgCl_2_·6H_2_O and Ca(OH)_2_. The FTIR results confirm the (B) position of carbonate and Rietveld method was applied to evaluate that Mg^2+^ substitutes essentially in the Ca (1) site; at the same time, the presence of Na^+^ ions was not discussed in the paper but could be assumed as a trace element [73].

Interestingly, at DFT level of calculations the incorporation of CO_3_^2−^ into the B position occurs as least favorable, irrespective of the scheme for the charge compensation used [67].

## 4. Conclusions

The results of our calculations at the DFT level revealed that the co-doping of HA by CO_3_^2−^ and Mg^2+^ can be more efficient than the doping of sole carbonate or magnesium ions. This is due to the energy gain from direct interactions between CO_3_^2−^ and Mg^2+^ ions. The complex is formed more eagerly in the tight anion channel, namely when CO_3_^2−^ replaces hydroxide ions, and Mg^2+^ resides in one of the Ca(2) positions close to the carbonate. When carbonate replaces PO_4_^3−^ ions, its interactions with Mg^2+^ residing in the proximity, at either Ca(1) or Ca(2) positions, are not beneficial. The theoretical results support the change in the FTIR spectra, published earlier, showing the increase in the carbonate population in the A position when Mg^2+^ is co-doped. The obtained results contribute to the understating of the doped HA structure and reveal the energy efficiency of a paired carbonate magnesium incorporation. Altogether, it paves the way to rational designs of bio-based catalytic systems.

## Figures and Tables

**Figure 1 materials-15-09046-f001:**
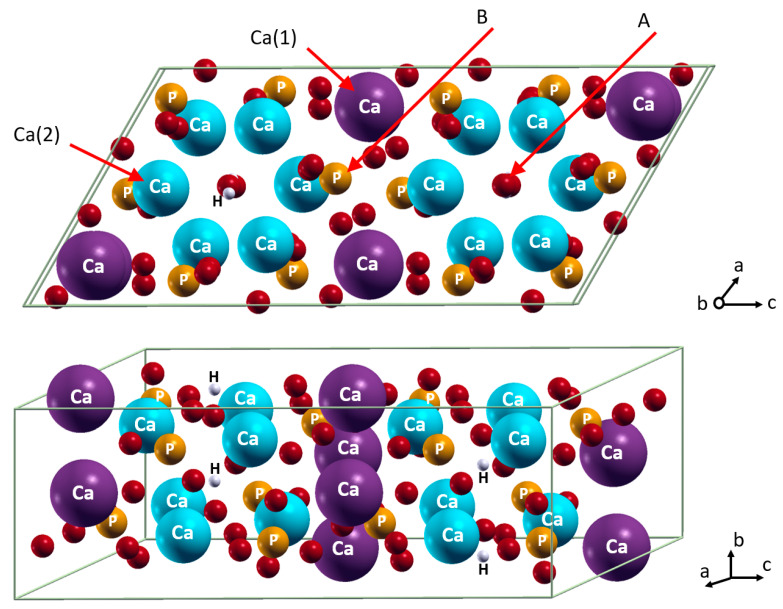
Incorporation of foreign cations in Ca(1) and Ca(2) positions and anions in A and B positions. Color coding: Ca in Ca(1) position–magenta; Ca in Ca(2) position–cyan; P–orange; O–red; H–white. For clarity, all atoms except oxygen are named.

**Figure 2 materials-15-09046-f002:**
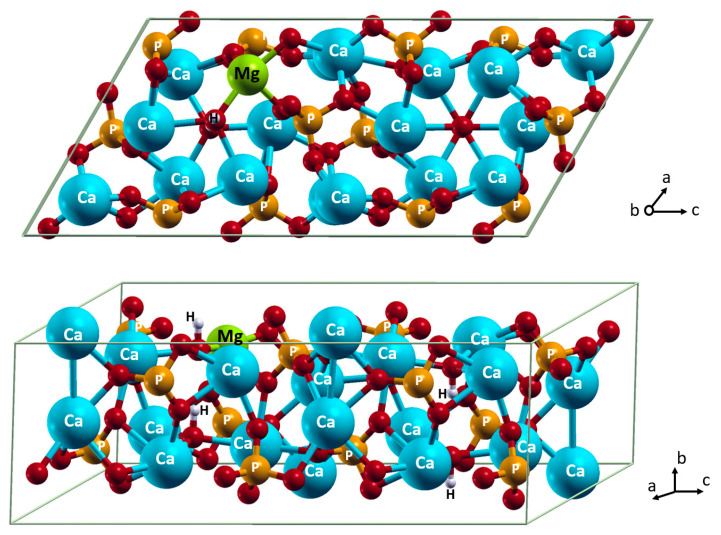
Incorporation of Mg^2+^ in Ca(2) position. Color coding: Ca–cyan; P–orange; O–red; H–white; Mg–light green. For clarity, all atoms except oxygen are named.

**Figure 3 materials-15-09046-f003:**
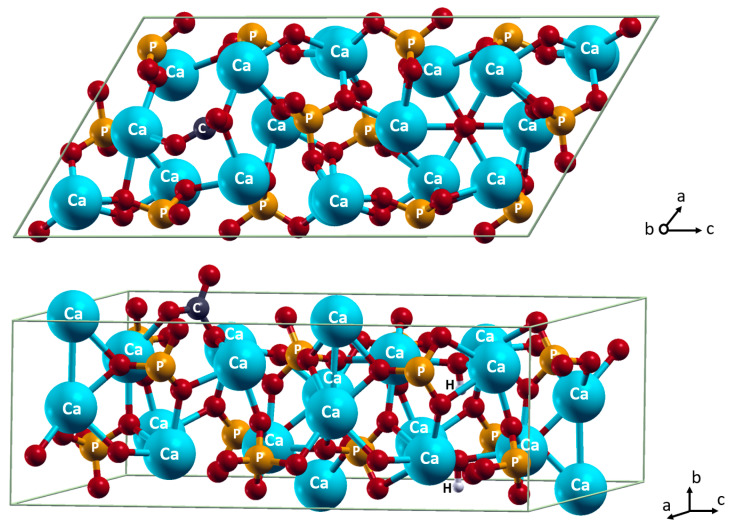
Incorporation of CO_3_^2−^ in A position. Color coding: Ca–cyan; P–orange; O–red; H–white; C–dark grey. For clarity, all atoms except oxygen are named.

**Figure 4 materials-15-09046-f004:**
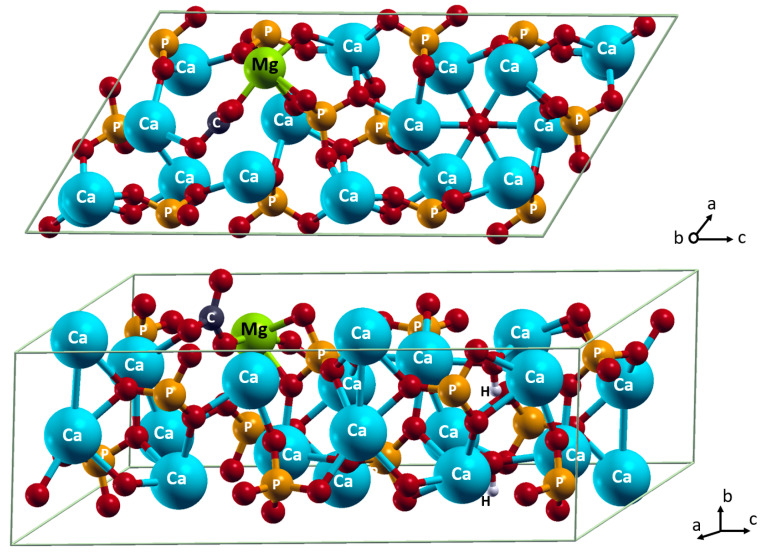
Incorporation of Mg^2+^ in Ca(2) position and CO_3_^2−^ in A position. Color coding: Ca–cyan; P–orange; O–red; H–white; C–dark grey; Mg–light green. For clarity, all atoms except oxygen are named.

**Table 1 materials-15-09046-t001:** Some parameters of the HA cell with formula Ca_10_(PO_4_)_6_(OH)_2_ and the cell with Mg^2+^ incorporation according to the formula Ca_9,5_Mg_0.5_(PO_4_)_6_(OH)_2_. The relative Ca–O bonds in pure HA are represented.

Incorporation	ΔE, eV	Cell Volume, Å^3^	Mg–O, Å	Ca–O, Å
HA		1068.2		
Mg in Ca(1)	0.042	1056.9	2.19	2.48
			2.19	2.42
			2.16	2.39
			2.22	2.45
			2.23	2.61
			3.02	2.52
Mg in Ca(2)	0	1059.2	2.09	2.38
			2.12	2.41
			2.21	2.55
			2.1	2.36
			2.52	2.37
			2.97	2.73
			3.75	3.49

**Table 2 materials-15-09046-t002:** Some parameters of the HA cell with CO_3_^2−^ incorporation with formula Ca_9.5_Na_0.5_(CO_3_)_0.5_(PO_4_)_5.5_(OH)_2_ and Ca_10_(CO_3_)_0.5_(PO_4_)_6_(OH)_1_, and the cell with Mg^2+^ incorporation according to the formula Ca_9_Na_0.5_Mg_0.5_(CO_3_)_0.5_(PO_4_)_5.5_(OH)_2_ and Ca_9.5_Mg_0.5_(CO_3_)_0.5_(PO_4_)_6_(OH)_1_. The relative Ca–O bonds in pure HA are represented.

Incorporation	ΔE, eV	Cell Volume, Å^3^	Mg–CO_3_^2−^, Å	Na–CO_3_^2−^, Å
CO_3_^2−^ in A	-	1082.2	-	-
Mg in Ca(1)	−0.005	1072.0	5.07	-
Mg in Ca(2)	−0.308	1082.6	1.98	-
CO_3_^2−^ in B, Na in Ca(1)	-	1062.2	-	4.08
Mg in Ca(1)	−0.076	1052.6	2.15	4.06
Mg in Ca(2)	−0.037	1056.2	2.25	3.87
CO_3_^2−^ in B, Na in Ca(2)	-	1060.2	-	2.86
Mg in Ca(1)	−0.055	1049.4	2.31	2.74
Mg in Ca(2)	−0.045	1053.1	2.12	2.92

## Data Availability

Not applicable.

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
