# Peer review of "The Mutual Incorporation of Mg2+ and CO32− into Hydroxyapatite: A DFT Study"

_materials, 2022, doi:10.3390/ma15249046_

Round 1

Reviewer 1 Report

Incorporation or replacement of Mg and CO2 in hydroxyapatite increase the efficiency and may be control of bioactivity because the contraction will increase the surface area and its effects . I have some comments :

1. It the beginning by the abbreviation of density functional theory DFT which must  mentioned in the first appearance 

2. The figures of hydroxyapatite  structure( coloured atoms ) , I would like you put the  atoms such as Ca , P, O, H inside    even it will be clear for black and white 

3. Little improving for the English Language is needed

Author Response

On behalf of the authors, I would like to express my heartfelt thanks to the reviewer for the valuable comments, which led to the improvement of our work. We checked the paper and carefully corrected it according to the recommendations.

Incorporation or replacement of Mg and CO2 in hydroxyapatite increase the efficiency and may be control of bioactivity because the contraction will increase the surface area and its effects . I have some comments :

  1. It the beginning by the abbreviation of density functional theory DFT which must  mentioned in the first appearance 

We removed this abbreviation from the Abstract. Now, the first mention of density functional theory in the main text of the article is accompanied by the abbreviation (DFT) L59

  1. The figures of hydroxyapatite  structure( coloured atoms ) , I would like you put the  atoms such as Ca , P, O, H inside    even it will be clear for black and white 

The names of atoms are indicated in the Figures, as suggested. The corresponding note is added to the captures “For clarity, all atoms except oxygen are named”.

  1. Little improvement for the English Language is needed

We carefully read out the text to enhance the stylistics.

Reviewer 2 Report

This paper is a study of The mutual incorporation of Mg2+ and CO32- into hydroxyapatite: a DFT study.

 The paper is well written, and the English is good.

 1. In the “Introduction, please explain or complete the sentence: The Mg...O distance in the complex is 1.98 Å (row 19).

2. In the “ Results and discussion” part of  3.1. Mg2+ incorporation into HA, please explain or complete the sentence Despite the most pronounced shortening of the Mg2+...O distance... (row 175).

3. In the “ Results and discussion” part of 3.2.1. Carbonate in the A position, same thing, explain or find another notation.

 I recommend a minor revision.

Author Response

On behalf of the authors, I would like to express my heartfelt thanks to the reviewer for the valuable comments, which led to the improvement of our work. We checked the paper and carefully corrected it according to the recommendations.

This paper is a study of The mutual incorporation of Mg2+ and CO32- into hydroxyapatite: a DFT study.

 The paper is well written, and the English is good.

  1. In the “Introduction, please explain or complete the sentence: The Mg...O distance in the complex is 1.98 Å (row 19).

The sentence was re-shaped to clarify the message: “In the most energy favorable complex, the distance between Mg2+ and O atom of carbonate ion decreased compared to Mg…O distances to the surrounding phosphate or hydroxyl ions, and amounted to 1.98 Å”.

  1. In the “ Results and discussion” part of  „1. Mg2+ incorporation into HA, please explain or complete the sentence Despite the most pronounced shortening of the Mg2+...O distance... (row 175).

We tried to make the message clearer: “One might anticipate that a more noticeable local condensation of ions would lead to a greater compression of the lattice cell. In contrast, despite the most pronounced shortening of the Mg2+…O distances is in Ca(2) position, the contraction of the cell lattice is more pronounced when Mg2+ is in Ca(1) position.”

  1. In the “ Results and discussion” part of2.1. Carbonate in the A position, same thing, explain or find another notation.

The conventional notation is explained in the Introduction and presented graphically in Figure 1. To recall it in the Results section we changed the relative subheadings (even the information seems redundant): “Carbonate in the OH (A) position” and “Carbonate in the PO43– (B) position”

 I recommend a minor revision.

Reviewer 3 Report

The following typos should be corrected.

Line 69 "calming"  should be claiming,

line 70 "reviling" should be revealing,

line 75 "Ca+"  should be Ca++,

line 111 "pronounceable" should be predictable,

line 245 "doing" should be doping.

Author Response

On behalf of the authors, I would like to express my heartfelt thanks to the reviewer for the valuable comments, which led to the improvement of our work. We checked the paper and carefully corrected it according to the recommendations.

The following typos should be corrected.

Line 69 "calming"  should be claiming,

Corrected

line 70 "reviling" should be revealing,

Corrected

line 75 "Ca+"  should be Ca++,

Corrected

line 111 "pronounceable" should be predictable,

Corrected

line 245 "doing" should be doping.

Corrected

Dr. Margarita Goldberg

Reviewer 4 Report

The authors studied hydroxyapatite doped with Mg and carbonate ions by DFT. The reviewer considers that this paper is suitable for this journal and many readers are interested in this paper. However, the reviewer feels that the contrast with similar studies is insufficient. To improve this paper, the reviewer asks the authors to modify this paper at the following points.

1.

Many such materials are being experimentally studied as potential biomaterials. Therefore, the reviewer would like the authors to explain in detail how this research will help us understand the physicochemical and biological properties of the material.

2.

The following papers are DFT studies of carbonated hydroxyapatite.

The reviewer requires to the authors to discuss the differences of these studies and your study and explain what is new finding.

https://doi.org/10.1016/j.actbio.2014.05.007

3.

The following papers are DFT studies of hydroxyapatite containing Mg ions.

The reviewer requires to the authors to discuss the differences of these studies and your study and explain what is new finding.

https://doi.org/10.1016/j.biomaterials.2010.11.017

4.

The following papers are DFT studies of the carbonated hydroxyapatite containing the divalent cations. The compositions of the target for DFT calculation are quite similar to this paper.

The reviewer requires to the authors to refer the paper and to discuss the differences of these studies and your study and to explain what is new finding.

https://doi.org/10.1111/jace.17263

The other comment:

Hydroxide ion (OH-) is described as hydroxyl ion, but this may be an error, so please check it out.

Author Response

Dear anonymous reviewer, we are very appreciative of your valuable comments. We corrected the manuscript and presented the answers below:

1.

Many such materials are being experimentally studied as potential biomaterials. Therefore, the reviewer would like the authors to explain in detail how this research will help us understand the physicochemical and biological properties of the material.

Thank you for your recommendation. The presented paper is the part of the cycle of the magnesium-doped hydroxyapatite investigations. The previous paper ("The improved textural properties, thermal stability, and cytocompatibility of mesoporous hydroxyapatite by Mg2+ doping." Materials Chemistry and Physics DOI: 10.1016/j.matchemphys.2022.126461) deals with the physicochemical and biological properties of the Mg2+ and CO32- dopped hydroxyapatite materials. We discussed in detail the influence of the Mg2+ concentration and CO32- incorporations on the phase composition, morphology, thermal stability, cytotoxicity and cytocompatibility of the hydroxyapatite materials. The presented paper deals with the theoretical investigations, thus the experimental influence of the incorporation ions presented in the introduction. At the same time, according to your recommendations, the influence of the magnesium and carbonate ions was described in more detail and the following references were added:

  1. Karunakaran, G.; Cho, E.B.; Kumar, G.S.; Kolesnikov, E.; Janarthanan, G.; Pillai, M.M.; Rajendran, S.; Boobalan, S.; Sudha, K.G.; Rajeshkumar, M.P. Mesoporous Mg-Doped Hydroxyapatite Nanorods Prepared from Bio-Waste Blue Mussel Shells for Implant Applications. Ceram Int 2020, 46, 28514–28527, doi:10.1016/J.CERAMINT.2020.08.009.
  2. Sader, M.S.; Lewis, K.; Soares, G.A.; LeGeros, R.Z. Simultaneous Incorporation of Magnesium and Carbonate in Apatite: Effect on Physico-Chemical Properties. Materials Research 2013, 16, 779–784, doi:10.1590/S1516-14392013005000046.
  3. Sprio, S.; Pezzotti, G.; Celotti, G.; Landi, E.; Tampieri, A. Raman and Cathodoluminescence Spectroscopies of Magnesium-Substituted Hydroxyapatite Powders. J Mater Res 2005, 20, 1009–1016, doi:10.1557/JMR.2005.0132.

2.

The following papers are DFT studies of carbonated hydroxyapatite.

The reviewer requires to the authors to discuss the differences of these studies and your study and explain what is new finding.

We extended the discussion in the section 3.2.2, as suggested by the Reviewer and also based on the following articles:

  1. Saito, T.; Yokoi, T.; Nakamura, A.; Matsunaga, K. Formation Energies and Site Preference of Substitutional Divalent Cations in Carbonated Apatite. Journal of the American Ceramic Society 2020, 103, 5354–5364, doi:10.1111/JACE.17263.
  2. Boivin, G.; Meunier, A.P.J. The Mineralization of Bone Tissue: A Forgotten Dimension in Osteoporosis Research., doi:10.1007/s00198-002-1347-2.
  3. Karaaslan, F.; Mutlu, M.; Mermerkaya, M.U.; Karaoǧlu, S.; Saçmaci, Åž.; Kartal, Åž. Comparison of Bone Tissue Trace-Element Concentrations and Mineral Density in Osteoporotic Femoral Neck Fractures and Osteoarthritis. Clin Interv Aging 2014, 9, 1375, doi:10.2147/CIA.S66354.
  4. Burnell, J.M.; Teubner, E.J.; Miller, A.G. Normal Maturational Changes in Bone Matrix, Mineral, and Crystal Size in the Rat. Calcified Tissue International 1980 31:1 1980, 31, 13–19, doi:10.1007/BF02407162.
  5. Rey, C.; Renugopalakrishman, V.; Collins, B.; Glimcher, M.J. Fourier Transform Infrared Spectroscopic Study of the Carbonate Ions in Bone Mineral during Aging. Calcified Tissue International 1991 49:4 1991, 49, 251–258, doi:10.1007/BF02556214.
  6. Rey, C.; Collins, B.; Goehl, T.; Dickson, I.R.; Glimcher, M.J. The Carbonate Environment in Bone Mineral: A Resolution-Enhanced Fourier Transform Infrared Spectroscopy Study. Calcified Tissue International 1989 45:3 1989, 45, 157–164, doi:10.1007/BF02556059.
  7. Tampieri, A.; Celotti, G.; Landi, E. From Biomimetic Apatites to Biologically Inspired Composites. Anal Bioanal Chem 2005, 381, 568–576, doi:10.1007/S00216-004-2943-0/FIGURES/11.

3.

The following papers are DFT studies of hydroxyapatite containing Mg ions.

The reviewer requires to the authors to discuss the differences of these studies and your study and explain what is new finding.

We discussed more thoroughly what is known up till now and emphasized the contradictory information on the Mg2+ positioning in HA from the literature. We also grouped the energy values estimated at the DFT level for easier navigation.

  1. Ren, F.; Lu, X.; biomedical, Y.L.-J. of the mechanical behavior of; 2013, undefined Ab Initio Simulation on the Crystal Structure and Elastic Properties of Carbonated Apatite. ElsevierPaperpile.
  2. Laurencin, D.; Almora-Barrios, N.; de Leeuw, N.H.; Gervais, C.; Bonhomme, C.; Mauri, F.; Chrzanowski, W.; Knowles, J.C.; Newport, R.J.; Wong, A.; et al. Magnesium Incorporation into Hydroxyapatite. Biomaterials 2011, 32, 1826–1837, doi:10.1016/J.BIOMATERIALS.2010.11.017.
  3. Andrés, N.C.; D’Elía, N.L.; Ruso, J.M.; Campelo, A.E.; Massheimer, V.L.; Messina, P. v. Manipulation of Mg2+-Ca2+ Switch on the Development of Bone Mimetic Hydroxyapatite. ACS Appl Mater Interfaces 2017, 9, 15698–15710, doi:10.1021/ACSAMI.7B02241/SUPPL_FILE/AM7B02241_SI_001.PDF.
  4. Saito, T.; Yokoi, T.; Nakamura, A.; Matsunaga, K. Formation Energies and Site Preference of Substitutional Divalent Cations in Carbonated Apatite. Journal of the American Ceramic Society 2020, 103, 5354–5364, doi:10.1111/JACE.17263.

4.

The following papers are DFT studies of the carbonated hydroxyapatite containing the divalent cations. The compositions of the target for DFT calculation are quite similar to this paper.

The reviewer requires to the authors to refer the paper and to discuss the differences of these studies and your study and to explain what is new finding.

Indeed, the authors of the mentioned article theoretically studied the efficiency of mutual inclusion for a number of divalent ions and carbonate ions. Our study is more focused on Mg2+ and CO32- interactions and not being contradicting the results of the Saito et al. describe in more detail the influence of such incorporation on local reorganization and cell geometry. The latter is important to benchmark the theoretical results against the experimental parameters, which, is turn, strongly dependent on the sintering conditions. Furthermore, in our calculations the charge compensation scheme differed from that used by Saito et al. Upon substitution of PO43- by CO32-, Saito et al. introduced H+ to the neighboring PO43− ion, resulting in the formation of HPO42− and a vacation of Ca2+. To less disturb the cell parameters, in our simulations, we used a less invasive scheme of charge compensation. Namely, the substitution of PO43- by CO32- was accompanied by the substitution of a Ca2+ by Na+. The substitution scheme realized upon synthesis would be crucial for HA-based catalytical systems, since the presence of HPO43− ions will increase acidic sites, but the addition of Na+ causes a moderate increase in the basicity of HAp. Furthermore, despite Na+ inclusion revealing no influence on the Mg2+ - CO32- complexation, such a result is of potential interest for the HA materials of biological origin.

The other comment:

Hydroxide ion (OH-) is described as a hydroxyl ion, but this may be an error, so please check it out.

Thank you, the “hydroxyl” was changed to “hydroxide”.

Reviewer 5 Report

The manuscript entitled “ The mutual incorporation of Mg2+ and CO32- into hydroxyapatite: a DFT study”.

 This article deals with DFT studies of Mg,CO3 co-substituted hydroxyapatite.

Although the objectives of this work seem to be interesting and the presented studies can be a good complement to typical experimental work, several parts of the manuscript should be improved and rewritten.

Therefore I cannot recommend this paper to be published as it is in Materials.

Specific comments:

ABSTRACT:

“ Hydroxyapatite (HA), an inorganic component of bones, has a stoichiometry composition Ca10(PO4)6(OH)2” – the Authors should emphasize that the biological apatites are not stoichiometric and that the Mg2+ and carbonate ions are the main impurities.

 INTRODUCTION:

·       The introduction lacks the most important thing - what are these studies for? How does the research fit in with the results obtained so far in other research centers? ·       There is only one reference to experimental research on co-substituted apatite - there have been more such studies and they should also be discussed. ·       Line 49 ---- Can silicate ions really build into the position of OH groups? ·       The word “inclusion” – should be precised.

RESULTS AND DISCUSSION:

·       The results are quite surprising to me. In bone apatite, carbonate ions are located mainly in the B position, although magnesium ions are also present in the structure. Authors should discuss their results with data from the literature on biological apatites (osseous and dental apatites). ·       The authors should also pay attention to the fact that in addition to the substitution of ions into the crystal structure, adsorption of ions on the crystal surface (both carbonate and magnesium ions) may occur. ·       It is not clear why the Authors in the introduction and abstract described that they were investigating the substitution of carbonate and magnesium ions, while data on sodium ions also appear in the results.  

OTHERS:

·       I believe that more attention should be paid to the preparation of citations – currently, they are inconsistent, sometimes the title of the article is missing, and sometimes the title of the journal.

Author Response

Dear anonymous reviewer, we very much appreciate your valuable comments. We corrected our manuscript and presented the answers below:

ABSTRACT:

“ Hydroxyapatite (HA), an inorganic component of bones, has a stoichiometry composition Ca10(PO4)6(OH)2” – the Authors should emphasize that the biological apatites are not stoichiometric and that the Mg2+ and carbonate ions are the main impurities.

Since we are restricted by 200 words in the abstract, we rephrased this sentence trying to be focused more on the application of non-natural HA. This would sound more consistent since our group is mainly focused on the controlled synthesis of hydroxyapatites in vitro.

The presence of Mg2+ and CO32- as main impurities in natural materials is outlined in the introduction.

INTRODUCTION:

The introduction lacks the most important thing - what are these studies for?

The presented paper is part of the cycle of the substituted HA "The improved textural properties, thermal stability, and cytocompatibility of mesoporous hydroxyapatite by Mg2+ doping." Materials Chemistry and Physics DOI: 10.1016/j.matchemphys.2022.126461. Searching the information in the literature we faced contradictory information about Mg2+ incorporation and little information about mutual Mg2+/CO32- ions. On the other hand, the data point to the fact that the incorporation of carbonate ions strongly depends on the method of synthesis used. This raised questions about how to interpret own experimental data. Furthermore, since the controlled synthesis of HA in vitro is a promising way to fabricate catalysts, to which the acid/basis ratio is of importance, the charge compensation scheme is crucial. In the current study, we described in detail the energy and cell parameters of the Mg2+/CO32- doped HA, which facilitates the control of the newly synthesized nanomaterials.

The aim is now stated in the text as follows: The positions of the foreign ions incorporation and the scheme of charge compensation would crucially affect the local acid/base properties of doped HA and their use in catalysis. The results of modeling would create a platform for the rational design of such materials and provide important parameters, as cell dimensions, to verify the ion's inclusion at a given experiment conditions.

How does the research fit in with the results obtained so far in other research centers? ·       There is only one reference to experimental research on co-substituted apatite - there have been more such studies and they should also be discussed. ·       

We extended the Introduction by discussing the results obtained by other groups, including:

  1. Karunakaran, G.; Cho, E.B.; Kumar, G.S.; Kolesnikov, E.; Janarthanan, G.; Pillai, M.M.; Rajendran, S.; Boobalan, S.; Sudha, K.G.; Rajeshkumar, M.P. Mesoporous Mg-Doped Hydroxyapatite Nanorods Prepared from Bio-Waste Blue Mussel Shells for Implant Applications. Ceram Int 2020, 46, 28514–28527, doi:10.1016/J.CERAMINT.2020.08.009.
  2. Sader, M.S.; Lewis, K.; Soares, G.A.; LeGeros, R.Z. Simultaneous Incorporation of Magnesium and Carbonate in Apatite: Effect on Physico-Chemical Properties. Materials Research 2013, 16, 779–784, doi:10.1590/S1516-14392013005000046.
  3. Sprio, S.; Pezzotti, G.; Celotti, G.; Landi, E.; Tampieri, A. Raman and Cathodoluminescence Spectroscopies of Magnesium-Substituted Hydroxyapatite Powders. J Mater Res 2005, 20, 1009–1016, doi:10.1557/JMR.2005.0132.
  4. Saito, T.; Yokoi, T.; Nakamura, A.; Matsunaga, K. Formation Energies and Site Preference of Substitutional Divalent Cations in Carbonated Apatite. Journal of the American Ceramic Society 2020, 103, 5354–5364, doi:10.1111/JACE.17263.
  5. Boivin, G.; Meunier, A.P.J. The Mineralization of Bone Tissue: A Forgotten Dimension in Osteoporosis Research., doi:10.1007/s00198-002-1347-2.
  6. Karaaslan, F.; Mutlu, M.; Mermerkaya, M.U.; Karaoǧlu, S.; Saçmaci, Åž.; Kartal, Åž. Comparison of Bone Tissue Trace-Element Concentrations and Mineral Density in Osteoporotic Femoral Neck Fractures and Osteoarthritis. Clin Interv Aging 2014, 9, 1375, doi:10.2147/CIA.S66354.
  7. Burnell, J.M.; Teubner, E.J.; Miller, A.G. Normal Maturational Changes in Bone Matrix, Mineral, and Crystal Size in the Rat. Calcified Tissue International 1980 31:1 1980, 31, 13–19, doi:10.1007/BF02407162.
  8. Rey, C.; Renugopalakrishman, V.; Collins, B.; Glimcher, M.J. Fourier Transform Infrared Spectroscopic Study of the Carbonate Ions in Bone Mineral during Aging. Calcified Tissue International 1991 49:4 1991, 49, 251–258, doi:10.1007/BF02556214.
  9. Rey, C.; Collins, B.; Goehl, T.; Dickson, I.R.; Glimcher, M.J. The Carbonate Environment in Bone Mineral: A Resolution-Enhanced Fourier Transform Infrared Spectroscopy Study. Calcified Tissue International 1989 45:3 1989, 45, 157–164, doi:10.1007/BF02556059.
  10. Tampieri, A.; Celotti, G.; Landi, E. From Biomimetic Apatites to Biologically Inspired Composites. Anal Bioanal Chem 2005, 381, 568–576, doi:10.1007/S00216-004-2943-0/FIGURES/11.

Line 49 ---- Can silicate ions really build into the position of OH groups? ·       

Anions, like CO32-, SiO44-, F-, and Cl-, can substitute anion positions, which in HA are PO43- (B-position) or OH- (A-position).

The word “inclusion” – should be precised.

We tried to use “incorporation” instead of “inclusion”

RESULTS AND DISCUSSION:

  • The results are quite surprising to me. In bone apatite, carbonate ions are located mainly in the B position, although magnesium ions are also present in the structure. Authors should discuss their results with data from the literature on biological apatites (osseous and dental apatites). ·

Additional information was added as well as previous data discussed.

  1. Karaaslan, F.; Mutlu, M.; Mermerkaya, M.U.; Karaoǧlu, S.; Saçmaci, Åž.; Kartal, Åž. Comparison of Bone Tissue Trace-Element Concentrations and Mineral Density in Osteoporotic Femoral Neck Fractures and Osteoarthritis. Clin Interv Aging 2014, 9, 1375, doi:10.2147/CIA.S66354.
  2. Burnell, J.M.; Teubner, E.J.; Miller, A.G. Normal Maturational Changes in Bone Matrix, Mineral, and Crystal Size in the Rat. Calcified Tissue International 1980 31:1 1980, 31, 13–19, doi:10.1007/BF02407162.
  3. Rey, C.; Renugopalakrishman, V.; Collins, B.; Glimcher, M.J. Fourier Transform Infrared Spectroscopic Study of the Carbonate Ions in Bone Mineral during Aging. Calcified Tissue International 1991 49:4 1991, 49, 251–258, doi:10.1007/BF02556214.
  4. Rey, C.; Collins, B.; Goehl, T.; Dickson, I.R.; Glimcher, M.J. The Carbonate Environment in Bone Mineral: A Resolution-Enhanced Fourier Transform Infrared Spectroscopy Study. Calcified Tissue International 1989 45:3 1989, 45, 157–164, doi:10.1007/BF02556059.
  5. Tampieri, A.; Celotti, G.; Landi, E. From Biomimetic Apatites to Biologically Inspired Composites. Anal Bioanal Chem 2005, 381, 568–576, doi:10.1007/S00216-004-2943-0/FIGURES/11.

   The authors should also pay attention to the fact that in addition to the substitution of ions into the crystal structure, the adsorption of ions on the crystal surface (both carbonate and magnesium ions) may occur. ·       

In our previous work we observed the introduction of adsorbed CO32- groups in the lattice during the heat -treatment according to FTIR data. The magnesium incorporation was confirmed by the lattice changes according to XRD results, which were demonstrated on the materials after the synthesis. Thus, in our paper, we assumed the incorporation of both ions.

It is not clear why the Authors in the introduction and abstract described that they were investigating the substitution of carbonate and magnesium ions, while data on sodium ions also appear in the results.  

The information is added to the Abstract.  

OTHERS:

  • I believe that more attention should be paid to the preparation of citations – currently, they are inconsistent, sometimes the title of the article is missing, and sometimes the title of the journal.

Thank you, we checked the citation list, the reference manager was corrected.

Round 2

Reviewer 5 Report

Thank you for addressing the comments. In the abstract section, the authors talk about the basic applications
of hydroxyapatite. However, they completely ignore its use as a bone substitute
and biomaterial. On the other hand, in the introduction they write about the role of bone apatite. As for my remark regarding SiO44- silicates - of course they are anions.
But it is worth checking whether they will actually fit in the column of hydroxyl
ions on the edge of the unit cell (their radius is quite large).
Please provide literature references that show that silicates can substitute OH groups. I still believe that the literature has not been prepared properly -
sometimes the authors use abbreviations of journal names, sometimes
full names. Sometimes there is no vol. number given, etc.

Author Response

Dear reviwer, thank you very much for all additional comments.

Finally, we found the data on the SiO44- substitution opportunity and corrected our text, as well as added the references on the B-type of the substitution. 

In the abstract we focused on the bone tissue substitution apllication.

The reference list was corrected by hands.

Thank you very much for inspiration to make our manuscript better!

Best regards, Dr. Margarita Gldberg on behalf of our authors.